# Sex Differences in the Frequencies of B and T Cell Subpopulations of Human Cord Blood

**DOI:** 10.3390/ijms241411511

**Published:** 2023-07-15

**Authors:** Michelle Bous, Charline Schmitt, Muriel Charlotte Hans, Regine Weber, Nasenien Nourkami-Tutdibi, Sebastian Tenbruck, Bashar Haj Hamoud, Gudrun Wagenpfeil, Elisabeth Kaiser, Erich-Franz Solomayer, Michael Zemlin, Sybelle Goedicke-Fritz

**Affiliations:** 1Department of General Pediatrics and Neonatology, Saarland University Medical Center, 66421 Homburg, Germany; michelle.bous@uks.eu (M.B.); muriel.hans@uni-saarland.de (M.C.H.); regine.weber@uks.eu (R.W.); nasenien.nourkami@uks.eu (N.N.-T.); elisabeth.kaiser@uks.eu (E.K.); michael.zemlin@uks.eu (M.Z.); 2Department of Gynaecology and Obstetrics, Saarland University Medical Center, 66421 Homburg, Germanybashar.hajhamoud@uks.eu (B.H.H.); erich.solomayer@uks.eu (E.-F.S.); 3Institute for Medical Biometry, Epidemiology and Medical Informatics (IMBEI), Saarland University, Campus Homburg, 66421 Homburg, Germany; gw@med-imbei.uni-saarland.de

**Keywords:** B cells, T cells, B1 cells, human, white blood cells, cord blood, development, ontogeny, sex differences, gender medicine

## Abstract

Cord blood represents a link between intrauterine and early extrauterine development. Cord blood cells map an important time frame in human immune imprinting processes. It is unknown whether the sex of the newborn affects the lymphocyte subpopulations in the cord blood. Nine B and twenty-one T cell subpopulations were characterized using flow cytometry in human cord blood from sixteen male and twenty-one female newborns, respectively. Except for transitional B cells and naïve B cells, frequencies of B cell counts across all subsets was higher in the cord blood of male newborns than in female newborns. The frequency of naïve thymus-negative Th cells was significantly higher in male cord blood, whereas the remaining T cell subpopulations showed a higher count in the cord blood of female newborns. Our study is the first revealing sex differences in the B and T cell subpopulations of human cord blood. These results indicate that sex might have a higher impact for the developing immune system, urging the need to expand research in this area.

## 1. Introduction

The development of the human immune system begins during intrauterine development and is not completed at birth [1]. Investigating sex differences in lymphocytes present in the cord blood essentially contributes to a deeper comprehension of the immune development in utero and of the potential consequences of these differences within the immune system. Although lymphocyte subpopulations evolve continuously and are influenced by environmental antigen exposure postpartum, analyzing the cord blood provides unique insights into the prenatal immune status and early immune imprinting before undergoing changes and adaptations.

The prenatal microenvironment contains information and signals that can shape immune development. These prenatal influences can affect the future reactivity, sensitivity, and functions of lymphocytes. The fetus learns to tolerate both self and maternal antigens, while simultaneously developing protective immunity in preparation for birth [2]. Epigenetic mechanisms, such as DNA methylation and histone demethylation, influence the gene expression patterns of lymphocytes during the prenatal stage and determine their future functionality [3,4,5].

After birth, many processes are initiated affecting the immune cells in particular due to changes in environmental antigen exposure [6,7]. Within the neonatal period, the immune system is immature and certain components of the adaptive immune response are not fully developed. Consequently, newborns are more prone to contracting infections due to their incomplete defense mechanisms against pathogens. The lack of immune memory leads to a limited protective immunity against infectious diseases, as they have only been exposed to a small range of pathogens [8]. In this context, viral and microbial colonization by perinatal transmission is of great importance in the maturation process of the innate and adaptive immune systems. Moreover, microbial dysbiosis within the first 100 days of life augments the risk of allergies during lifetime [9].

Moreover, there are notable differences in the antibody response of neonates when compared to adults. Although neonates benefit from maternal antibodies providing maternal passive immunity, their antibody titers are lower compared to adult blood [10,11].

During infancy, the immune system matures. As a continuous learning process, immunity is built up during pre-puberty by recurrent infections and vaccinations. With the onset of puberty, the immune system achieves similar functional capacities compared to a healthy, mature adult immune system [12].

The maturation of immune cells is a complex multi-layered process. Griffin et al. [13] described human B1 cells as a cluster of differentiation (CD) CD20+ CD27+ CD43+ B cell subpopulation based on the typical B1 functions observed in murine B1 cells: spontaneous immunoglobulin (Ig) IgM secretion, efficient T cell stimulation, and continuous intracellular signaling [13]. B1 cells have a huge percentage in the cord blood (CB) (50%) but decrease over the years [13,14].

Marginal zone B (MZ B cells) cells are non-circulating mature B cells which form an interface between blood and the lymph nodes [7,15]. Splenic MZ B cells, located in the marginal zone of the spleen, are part of the innate immune system responding to blood-borne antigens. MZ B cells only account for 5% of the innate and adaptive immune system; nevertheless, they are of utmost importance for early immune responses, interacting effectively with rapid antibody reactions on viral and bacterial pathogens [15].

Thymus-negative cells represent a premature stage of T helper (T_h_^−^) lymphocytes prior to the upcoming positive selection in the thymus. In the positive selection process, T_h_ cells can bind major histocompatibility complexes (MHC) to receive positive growing signals and enter the cell cycle. Following the positive selection process, cells that bind to self-antigens are induced to apoptosis [16]. These selection processes take place in the thymus postnatally, assuming that the maturation of T_h_ cells is not finished by the time of birth [17].

One important and long neglected fact are the differences between females and males. Investigating biological (sex) and sociocultural (gender) differences has gained more importance over the past few years. Within the field of medical research, most of the performed research and published data refers to male individuals. Sex differences have been found in certain diseases like asthma, diabetes, and cardiovascular disease. Women are at higher risk to develop diseases like Takotsubo cardiomyopathy, or show different manifestations of common diseases, like heart failure or myocardial infarction. Moreover, autoimmune diseases, such as systemic lupus erythematosus (SLE), multiple sclerosis (MS), rheumatoid arthritis, and Hashimoto’s thyroiditis are more common in women than in men [18,19].

To gain a deeper comprehension of the underlying causes of these sex-specific differences later in life, a profound investigation of these sex differences and their impact on early immune development is of utmost importance. Sex differences in immune responses can be mainly traced back to two main factors: the influence of sex hormones, such as testosterone and estrogen, and differences in the number of immune-related genes on the X chromosome [20]. The production of androgens in the male fetus begins within the first trimester at ten weeks of gestation, marking an early initiation of sex differentiation while also affecting the developing immune system [20].

There is a constant increase in the testosterone serum level in male newborns over the first three months after birth, with a decrease at 7–12 months of life, followed by a maintenance of low-level serum testosterone until the onset of puberty. In female newborns, there is a decrease in testosterone serum levels after birth. These sex differences are uniquely observed within the first year of life, as serum testosterone reaches equal levels in male and female children at pre-puberty, followed by a significant change with the onset of puberty [20]. Overall, estrogen stimulates the proliferation of B cells, while testosterone suppresses the latter [21]. Regarding T cell subpopulations, testosterone also suppresses early-stage lymphopoiesis. Passing several developmental steps, T cells abandon the expressing androgen receptors. Although female neutrophils and macrophages showed a better phagocyte activity [22], testosterone reduces the production of tumor necrosis factor (TNF) and expression of toll-like receptor 4 (TLR-4), and increases cytokine release. Therefore, autoimmune-mediated diseases are more frequently observed in women [20].

Alongside hormones, the X-chromosome also plays an important role in this context. Coding regions for genes involved in immune responses, such as toll-like receptors, cytokine receptors, and transcription factors can be found on the X-chromosome [23]. During embryogenesis, there is a random silencing of one X chromosome. At the same time, there is an escape out of this silencing process in 15% out of the immune-related genes; with 10% of these genes underlying the different levels of inactivation [24]. Thus, an overexpression of these immune-related genes is associated to X disomy. Genes associated to autoimmunity are also found on the X chromosomes, such as the expression of TLR-7, which has described in relation to lupus-associated autoimmunity [25].

A recent example for sex disparity and their associated different clinical courses and outcomes are infections with severe acute respiratory syndrome coronavirus 2. The underlying mechanism remains elusive, and the influence of estrogens and androgens, as well as androgen-sensitive genes coding for angiotensin-converting enzyme 2 (ACE2) and cell surface transmembrane protease serine 2 (TMPRSS2) are under discussion, as well as their sociocultural influences [26].

The cord blood reflects a unique period of early immune system development. Until now, there has been limited evidence regarding sex differences in the leucocytes in the cord blood. Our study aims to gain a deeper comprehension of the composition of the B and T cell subpopulations in the cord blood with respect to sex. In addition, we aimed to investigate whether there are sex-related differences within the lymphocyte subpopulations in the cord blood.

## 2. Results

### 2.1. B Cells

We analyzed the cord blood samples of 16 male newborns and 21 female newborns using flow cytometry. The mean of B1 cells was significantly higher in the cord blood of male neonates than in the cord blood of female neonates (male 3.7% ± 2.9 vs. female 1.8% ± 1.5, *p* = 0.023, Figure 1, Table 1). B1 cells are marked as CD20+ CD27+ CD43+ and are given as a percentage of the number of total CD20+ cells. Marginal zone memory B cells, marked as CD19+ CD27+ IgD+ IgM+, and given as a percentage of total CD19+ cells, were also found to be significantly higher in male cord blood (male 1.7% ± 2.1 vs. female 0.8% ± 1.2, *p* = 0.01, Figure 2, Table 1). With the exception of the naïve B cells and transitional B cells, the B cell count in all other subsets (innate B cells, class-switched memory B cells, late memory B cells, plasmablasts, transitional B cells, and B2 cells) was found to be higher in the cord blood of male newborns (Table 1).

### 2.2. T Cells

We analyzed T cell subpopulations in two panels. In T cell panel 1, we analyzed 16 male and 21 female newborns, and in T cell panel 2 we analyzed 11 male and 16 female newborns with flow cytometry, respectively.

Cytotoxic T cells, naïve effector T_h_ cells, naïve effector cytotoxic T cells, and naïve thymus-negative T_h_ cells all demonstrated an increasing trend in the cord blood of male newborns. Naïve thymus-negative T_h_ cells were significantly more abundant in male cord blood (male 24.4% ± 9.8 vs. female 15.9 ± 5.7, *p* = 0.05, Figure 3, Table 2). In cord blood, 12 other T cell subpopulations (T helper cells, T helper cells with αβ-TCR, activated cytotoxic T helper cells with αβ-TCR, memory effector T helper cells, naïve central T helper cells, naïve central T helper cells, memory effector cytotoxic T cells, memory central cytotoxic T cells, naïve central cytotoxic T cells, and naïve thymus-positive T helper cells) showed a non-significant higher count in the cord blood of female newborns than in male newborns (Table 2). The remaining subpopulations (T helper cells 1, naïve T helper cells 1, memory T helper cells 1, T helper cells 2, naïve T helper cells 2, memory T helper cells 2, and regulatory T cells) showed similar cell counts in both sexes.

## 3. Discussion

In our study, we found significant differences in the B and T cell populations (Table 3) between male and female newborns. We were able to generate reference values for the means of nine B cell subpopulations and twenty-one T cell subpopulations, respectively.

Although sex-related immunity differences have been observed in children and adults, very little is known about sex differences at birth. Research data about the latter are sparse, and pathophysiologic pathways and their mechanisms remain elusive. Sex differences in immune responses are traced back to two main factors: the influence of sex hormones, such as testosterone and estrogen, and differences in the number of immune-related genes on the X chromosome [20].

The production of androgens in the male fetus begins within the first trimester at ten weeks of gestation, marking an early initiation of sex differentiation, and also affecting the developing immune system [20]. In vitro, estradiol increases the accumulation of B cells, but in non-toxic doses, it does not affect the proliferation response. Moreover, it exhibits the same effect on the lymphocytes of both males and females. Non-toxic concentrations of testosterone do not influence the maturation of B cells in vitro [1]. Sex hormones also influence the T cell subpopulations. For instance, testosterone suppresses lymphopoiesis only during its early stages. After further development, T cells do not express androgen receptors. However, estrogen receptors can be expressed at any stage of T cell development: low doses of estrogen increase CD4+ cells, while high doses of estrogen reduce CD4 and CD8 cell counts [27]. In our analysis, the means of four out of fourteen T cell subpopulations (cytotoxic T cells, naïve effector T_h_ cells, naïve effector cytotoxic T cells, and naïve thymus-negative T_h_ cells) showed a trend to be higher in the cord blood of male newborns. The remaining T cell subpopulations were expected to be higher in the female newborns.

Here, the umbilical cord blood allows a special insight into the intrauterine lymphocyte profiles, as it represents an interphase between the pre–and postnatal physiology. In our analysis, B cell populations were higher in the male newborns compared to the female newborns, except for the naïve B cell and transitional B cell populations. These results are contrary to the relation of B cells in adult blood, with higher B cell counts observed in females than in males [28]. Possible reasons include changes in sex hormone levels in utero compared to levels after birth: estrogen stimulates the proliferation of B cells, while testosterone suppresses the latter [21]. After birth, there is an increase in testosterone levels in male newborns during the first three months, followed by a decrease to much lower levels before reaching pre-puberty low levels (at around 7–12 months of age). Females, on the other hand, experience a decrease in testosterone immediately after birth. Beyond the first year of life until adolescence, there are no sex-specific differences impacting the testosterone levels [20,29]. This fact may lead to the assumption that sex hormones are less relevant during this period of life. In this time, partially unknown factors, apart from sex hormones, might play an important role in the development of the immune system.

The influence of immune-related genes on the X chromosome, leading to sex differences in the development of immune-associated diseases, begins early in life [20].

X disomy leads to a higher genetic variety observed among these immune-related genes. Toll-like receptor 7/8 is able to detect viral single-stranded RNA (ss-RNA) and induce protective cytokine responses, in particular IFN responses. Although male and female innate immune cells do not differ significantly in terms of their overall TLR 7/8 expression, a humanized mouse model showed a positive influence of TLR7 ligation on the plasmacytoid dendritic cells and its interferon-α (IFN-α) and TNF responses [30]. Micro-RNAs (mi-RNA) as non-coding RNAs are associated with inflammatory diseases. The loci of mi-RNA have been found of higher amounts on the X chromosomes compared to the Y-chromosomes and autosomes [30]. X-linked miR223 seems to play a crucial role: studies have suggested that miR223-negative mice showed more inflammatory symptoms compared to non-deficient mice following *Candida albicans* infection, wherein miR223 is supposed to affect granulocyte generation and maturation negatively [31,32].

Men are more prone to infection-induced inflammation [33], like respiratory tract infections (RTI). Being more susceptible to RTI than women, severe courses of RTI infections are more often seen in males. This might be due to the innate immune response and its sex differences. A possible explanation could be the observed imbalance of TLR-2 and TLR-4: there is a higher expression of TLR-4 in the macrophages of male mice following endotoxic shock, leading to the extensive production of pro-inflammatory cytokines [34]. A protective factor seems to be the expression of TLR-2, which was found to be higher in female mice augmenting resistance against viral infections (especially coxsackie virus) [35]. Estrogen, on the other hand, suppresses lung inflammation in animal models [35]. The X-chromosome, and its linked immune-related genes, such as TLR 7/8-encoding genes, might also be responsible for these sex differences in infection-induced inflammation [36].

Former research has revealed a postnatal increase in CD19+ B cells and CD3+ T cells, until the age of 2 years, followed by a gradual decrease until adulthood. While CD3+ CD4+ T lymphocytes comply with this trend, the subpopulation of CD3+ CD8+ T lymphocytes remains stable for the first two years of life followed by a constant decrease until reaching levels similar to adults [37]. Overall, no sex differences were considered in the aforementioned studies. Various influences on the lymphocyte populations within this period, such as X chromosomal genes, must be considered when analyzing data on the developing immune system. There is less evidence available in the literature on sex-related changes in these lymphocyte subpopulations, and more research is necessary to illuminate all factors leading to the observed changes.

Even though there is a lot of data available on the CD4+ and CD8+ T cell sex differences in general, there is still a lack of evidence regarding sex differences in smaller lymphocyte subpopulations. We provided reference values for the means of 21 different T cell subpopulations. There is a need for further studies to gain more data about the sex-related differences in these subgroups in order to identify the sex-related risk factors for specific diseases.

Regardless of the sparse data on sex differences in the immunity (or immune development) of neonates, the published research so far indicates that male newborns are more prone to develop a robust innate immunity, characterized by higher counts of natural killer (NK) cells, monocytes, and basophils. Male newborns additionally show a better pro inflammatory response compared to females [23]. However, females do have higher CD4+ T cells and CD4/CD8 ratios, lower CD8+ T cells, and lower T_reg_ cell frequencies [23]. The numbers of B cells, IgG, and IgM are similar in both sexes, whilst male newborns show a higher IgA and IgE count than females [23]. Our data suggest that there are no changes in the total population of B or T cells, but there are changes in their subpopulations. The subpopulations of the B and T lymphocytes are very inhomogeneous groups that exert different influences and react differently to certain stimulations. More studies are needed to identify the factors leading to changes in these lymphocyte subpopulations [23].

Sex differences in in the immune response can be observed throughout childhood. Males exhibit higher levels of inflammation and NK cells, while CD4/CD8 ratios, CD8+ cells, CD4+ cells, and B cell numbers are similar in both sexes [23,33]. The onset of puberty marks a shift in the sex-specific immune response, potentially influenced by sex steroids. In adult females, higher inflammation, CD4/CD8 ratios, and CD4+ cell counts are observed, whereas adult males display higher counts of CD8+ cells, B cells, and immunoglobulins, with Treg cells also being higher in males [23]. After menopause, males exhibit a higher pro-inflammatory response, while females show an increased T cell activation. Other sex differences, such as the CD4+ CD8+ cell count and the CD4/CD8 ratio, remain equally distributed [23].

We found a significant difference in the MZ B cells (*p* = 0.025). Even though they only count for 5% of all B cells, they play an important role in the early immune response. MZ B cells are located at a strategic interface between blood and the lymph nodes, meaning they are able to react promptly with the antibody response and bacterial pathogens [1]. They close the gap between the early immune response and the late adaptive antibody reactions of follicular B cells [15]. This finding contradicts previously published data, as females of all ages have been found to exhibit greater total B cell numbers. The latter could be an example of changes in the composition of lymphocyte populations after birth, which may contribute to the sex-related differences observed in diseases during the neonatal period and later in life.

B1 cells, defined as CD20+ CD27+ CD43+, also exhibited a significant difference (*p* = 0.01). B1 cells constitute the main portion of B lymphocytes in newborns, but decrease to approximately 10% by adulthood [13].This illustrates that B1 cells are a dynamic component of the immune system, undergoing continuous changes during the development of a “mature” immune response. B1 cells play a crucial role in the innate immune system. However, certain underlying pathomechanisms remain elusive [38]. Establishing a consistent definition of immune markers is essential to ensure comparability among the studies investigating the B1 cell (sub)-populations. A major finding of our study was a higher count of B1 cells in male newborns. B1 cells are associated with the pathogenesis of asthma and allergies, particularly in relation to the production of IgE antibodies, which promote the regulation of the inflammatory processes [39,40], and the development of autoimmune diseases, predominantly affecting women [41]. Our findings suggest changes in the composition of B1 cells after birth, potentially due to environmental influences, antigen exposure, or the development of the adaptive immune response.

Thymus-negative T cells were found to be significantly higher in male newborns (*p* = 0.005). Innate T cells migrate from the bone marrow to the thymus, where they undergo both positive and negative selection. During the positive selection process, T_h_ cells that are able to bind to the MHC complexes receive a positive growing signal and enter the cell cycle. This positive selection takes place in the postnatal thymus, indicating that the maturation of T_h_ cells is still an ongoing process by the time of birth [1]. Thymus-negative T cells are innate cells that have not completed positive and negative selection. After negative selection, mature T cells leaving the thymus tissue usually carry the CD31 marker. Negative selection plays a crucial role in the prevention of the development of autoimmune reactive cells. There are tissue-specific antigens expressed in the thymus. Autoimmune regulator genes (AIRE) control the expression of these antigens. Mutations in this gene can result in an autoimmune disease known as autoimmune polyglandular syndrome 1 (APS-1) [16]. Therefore, changes in the thymus-negative T cell count could be involved in development of autoimmune diseases, including APS-1. There are studies hinting at a correlation between a failure in negative thymus selection and rheumatoid arthritis [42]. Alterations in thymus-negative cell counts could potentially induce dysregulation in the immune system, resulting in either a reduced or incorrect pathogen detection.

This research can potentially lead to improvements in prenatal diagnostics by introducing new early detection screenings during pregnancy, including tests capturing specific parameters related to sex-specific differences in lymphocyte subpopulations and other immunological features. A better understanding of early childhood immunity could also result in adjustments towards sex-specific vaccination schedules. In the clinical context, a sex-based medication regime is possible based on the different immunological mechanisms. Furthermore, examining the prenatal phenotype of lymphocytes could build a basis for the development of individually tailored therapeutic approaches.

There are several limitations in our study. The main limitation is the very small sample size. The latter may contribute to that fact that small differences might not be detected within our cohort. In addition, small cohorts may not be representative for the overall population, thereby limiting the generalizability of our findings. Overall, small cohorts restrict the range of statistical analyses that can be performed due to the limited number of cases. Nevertheless, pre- and perinatal variables regarding the recruited neonates and their mothers have been taken into account to avoid confounders. Only mothers without autoimmune diseases and without any infections or antibiotic intake within the last 2 weeks before birth were included. Furthermore, only infants delivered by caesarean section were included in the study to enhance the comparability. All included children were term-born (up to 37 + 0 weeks of gestation).

Further studies are necessary to validate these findings. Follow-up examinations to investigate the relationship between the gender-specific differences in the umbilical cord blood and the occurrence of diseases are feasible. To detect changes in immunological development, follow-up blood samples, optimally during routine examinations, are necessary. Additional investigations with an extended cohort size are needed to verify these results. Gene expression analysis or epigenetic studies must be considered in future research to investigate the underlying molecular mechanisms.

In conclusion, our study identified significant differences in cord blood lymphocytes related to male and female neonates. Sex-related differences undergo changes throughout the development towards a mature immune response. We provide reference values for nine B cell and 21 T cell subpopulations in the cord blood, suggesting valuable data on the influence of sex on the developing immune system. Further investigations are elusive to enlighten the understanding of early immune development and its sex-related differences in neonates, infants, children, and young adults, along with their clinical consequences.

## 4. Materials and Methods

### 4.1. Patient Samples

We collected blood samples from the umbilical cord blood of term neonates (range 37–40 weeks gestation, *n* = 37) without any infectious, immunologic, or chronical disease or antibiotic treatment in the previous two weeks, before planned caesarean sections. The Ethics Committee of Saarland approved the study protocol (198/20, 6 August 2020). The written informed consent of all the parents was obtained.

### 4.2. Cell Isolation, Storage, and Counting

#### Cell Isolation and Storage

Cord blood was collected into S-Monovette^®^ K3 ethylene diamine tetraacetic acid (EDTA, Sarstedt, Nümbrecht, Germany). Cord blood mononuclear cells (CBMCs) were prepared from the whole blood through density gradient centrifugation using the Lymphocyte Separation Medium 1.077, FicoLite-H (Linaris biological products, Dossenheim, Germany). Whole blood was diluted 1:1 with D-PBS followed by layering the mixture 1.5:1 on FicoLite-H. Samples were centrifuged at 400× *g* for 30 min. Interphase was considered as CBMCs and were separated. Cells were washed twice using Dulbecco’s phosphate-buffered saline (D-PBS, Sigma Aldrich, Steinheim, Germany) and were centrifuged at 400× *g* for 10 min. Subsequently, CBMCs were resuspended in fetal bovine serum (FBS, Thermo Fisher Scientific, Waltham, MA, USA) + 10% dimethylsufoxid (DSMO, Sigma Aldrich, Steinheim, Germany) and were placed in a CellCamper^®^ for cryopreservation at −80 °C.

### 4.3. Flow Cytometry

On the day of the experiment, the cryopreserved cells were gradually thawed by first acclimating the frozen vial on ice and then thawing the cell pellet in a water bath at room temperature. Cell pellets were then washed twice using D-PBS and centrifuged at 400× *g* for 10 min) prior to the quantification of the viable and dead cells. For this purpose, a portion of the cell suspension was stained with acridine orange and propidium iodide (AO/PI) and was quantitatively analyzed using the LUNA-FL™ Automated Fluorescence Cell Counter (Logos Biosystems, Dongan-gu Anyang-si, Gyeonggi-do, Republic of Korea) according to the manufacturer’s instructions. Following this, BD Horizon Fixable Viability Stain 780 (BD Biosciences, Heidelberg, Germany) was used for staining the dead cells according to the manufacturer’s instructions. Cells were washed using BD CellWASH™ (BD Biosciences, Heidelberg, Germany) and were centrifuged at 500× *g* for 10 min. Fluorochrome-conjugated antibodies are listed in Table 4. Cell pellets were resuspended in a panel-specific antibody mix containing BD Horizon™ Brilliant Stain Buffer (BD Biosciences, Heidelberg, Germany) and BD Pharmingen™ Stain buffer BSA. Samples were incubated protected from light at room temperature for 30 min. The remaining erythrocytes were lyzed with BD FACS™ Lysing Solution (BD Biosciences, Heidelberg, Germany) according to the manufacturer’s protocol. The cell suspension was diluted by BD CellWASH™ and was centrifuged at 500× *g* for 10 min. Cell pellets were resuspended in D-PBS and were then analyzed on a 3-laser, 12-color FACS Celesta flow cytometer (Becton, Dickinson and Company, Heidelberg, Germany).

Flow cytometer performance, stability, fluorescence calibrations, and data reproducibility were sustained and checked daily using the Cytometer Setup and Tracking Module (CS&T, BD Biosciences, Heidelberg, Germany). Application settings were applied for each acquisition and Sphero™ Rainbow Calibration Particles (8 Peaks, BD Biosciences, Heidelberg, Germany) were used to validate the reproducibility of the data over time. Compensation settings were calculated using BD™ CompBeads (BD Biosciences, Heidelberg, Germany). Comparison of the stained sample with an unstained sample, a sample subjected to all procedures except antibody staining, and the isotype control to specify the non-specific binding of the antibodies, constituted the basis for the positive staining and gating strategy. Cell aggregates were removed from the analysis (FSC-A/FSC-H) and dead cells were excluded from the analysis by staining with BD Horizon Fixable Viability Stain 780. Lymphocyte classification based on morphological parameters (FSC-A/SSC-A) was confirmed at the end.

Data were acquired using FACSDiva (BD Biosciences, Heidelberg, Germany) and were analyzed using FlowJo v10 (BD Biosciences). The gates were preset and the measurements were performed blinded for sample identity (Figure 4, Figure 5 and Figure 6).

### 4.4. Statistical Analysis

Statistical analysis was performed using SPSS 28.0.0. (IBM, Chicago, IL, USA). Quantitative data was examined for normal distribution using the Shapiro–Wilk test. The majority of variables did not follow a normal distribution. Thus, we used the exact version of the Mann–Whitney U test for all comparisons between the male and female newborns. The *p*-values were two-sided and subjected to a significance level of 0.05. Due to the explorative nature of the investigation, we did not account for multiple statistical testing, and therefore reported raw, unadjusted *p*-values. Descriptive statistics were reported as mean with standard deviation.

## Figures and Tables

**Figure 1 ijms-24-11511-f001:**
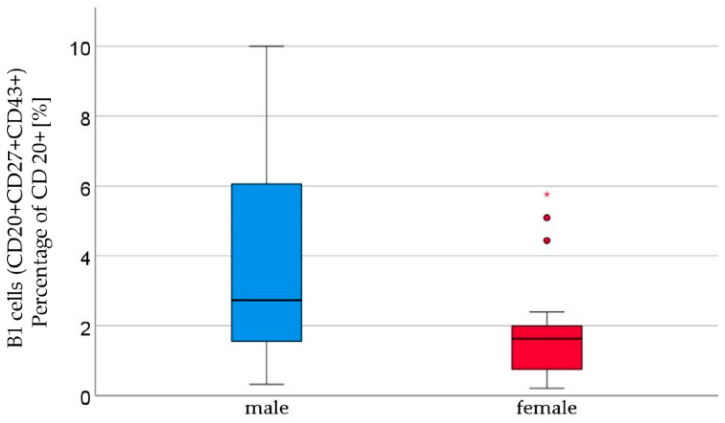
B1 lymphocytes (CD20+ CD27+ CD43+) percentage of CD20+ [%] were significantly higher in male (*n* = 16) than in female (*n* = 21) newborns, given with the median (male = 2.7, female = 1.6), minimum (male = 0.3, female = 0.2), and maximum values (male = 10, female = 5.8); mean ± standard deviation [%] (male = 3.7 ± 2.9, female = 1.8 ± 1.2). Outliers are marked with dots and (*).

**Figure 2 ijms-24-11511-f002:**
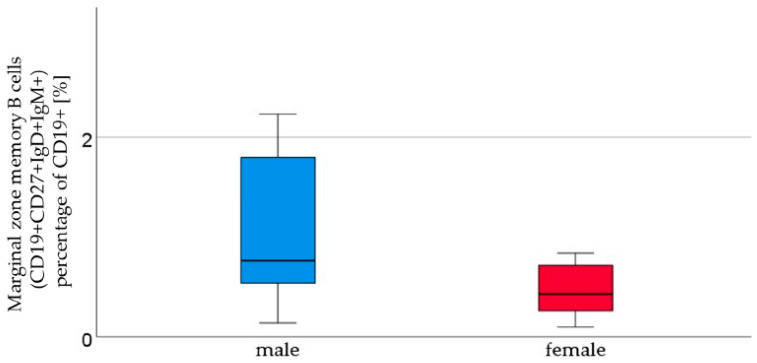
Marginal zone B lymphocytes (CD19+ CD27+ IgD+ IgM+). The percentage of CD19+ cells [%], were higher in male (*n* = 16) than in female (*n* = 21) newborns, given with the median (male = 0.8, female = 0.4), minimum (male = 0.1, female = 0.1), and maximum values (male = 7, female = 4.3); mean ± standard deviation [%] (male = 1.7 ± 2.1, female = 0.8 ± 1.2).

**Figure 3 ijms-24-11511-f003:**
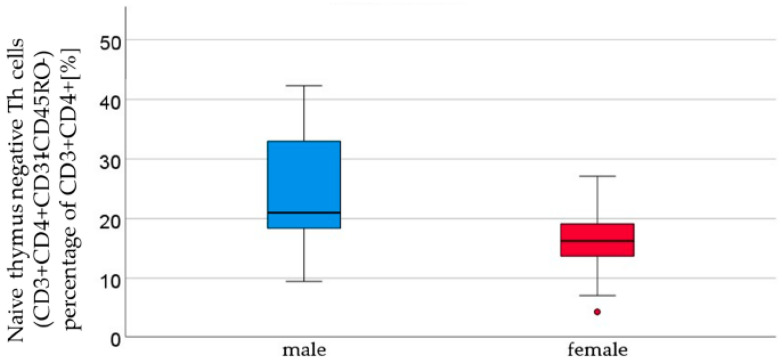
Thymus-negative T lymphocyte (CD3+ CD4+ CD31− CD45RO− ) percentages of CD3+ CD4+ [%] were significantly higher in male (*n* = 16) than in female (*n* = 21) newborns (see Figure 1), given with the median (male = 21, female = 16.2), minimum (male 9.4, female 4.2), and maximum values (male 42.3, female = 27.1); mean ± standard deviation [%] (male = 24.4 ± 9.8, female = 15.9 ± 5.7). Outliers are marked with dots.

**Figure 4 ijms-24-11511-f004:**
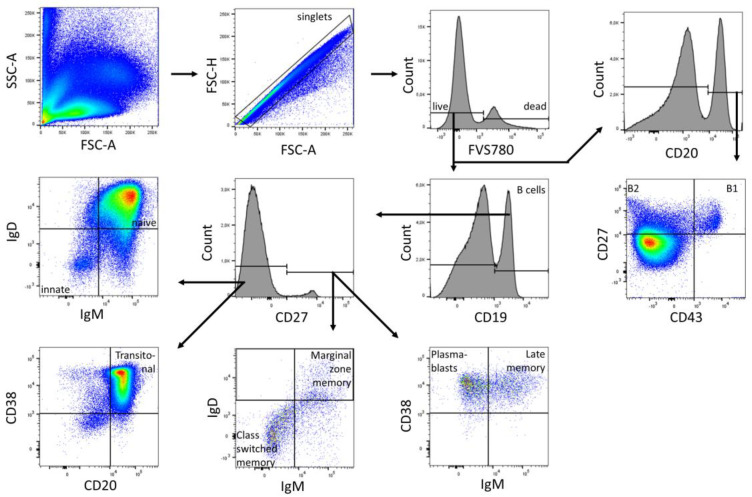
Gating strategy for B cell subpopulations of a representative CBMC sample. Analysis was performed in FlowJo (BD Biosciences). The gating strategy was based on morphological parameters (FSC-H/FSC-A). FVS 780-negative cells were considered as viable cells. From the viable cells, the number of CD19+ and CD20+ cells was determined. The CD19+ fraction was divided into CD27+ and CD27− B cells. CD27+ cells were subdivided into marginal zone memory B cells (IgD+ IgM+), class-switched memory B cells (IgD− IgM−), late memory B cells (CD38+ IgM+) and plasmablasts (CD38++ IgM−). CD27− cells were subdivided into innate B cells (IgD− IgM−), naïve B cells (IgD+ IgM+), and transitoneal B cells (CD20+ CD38+). CD20+ cells were subdivided into B1 cells (CD27+ CD43+) and B2 cells (CD27+ CD43−).

**Figure 5 ijms-24-11511-f005:**
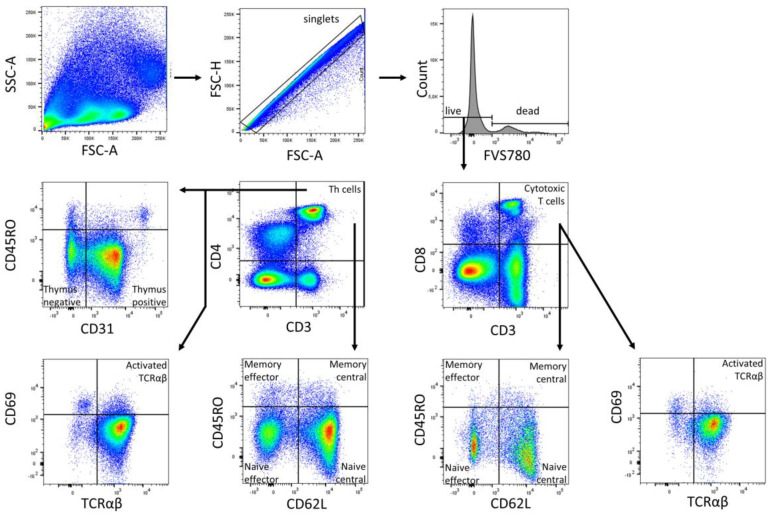
Gating strategy for T cell subpopulations (Panel 1) of a representative CBMC sample. Analysis was performed in FlowJo (BD Biosciences). The gating strategy was based on morphological parameters (FSC-H/FSC-A). FVS 780-negative cells were considered as viable cells. From the viable cells, the number of CD3+ cells was determined. The CD3+ fraction was divided into CD4+ T helper cells and CD8+ cytotoxic T cells. Both cell fractions were subdivided into naïve central (CD62L+ CD45RO−), naïve effector (CD62L− CD45RO−), memory effector (CD62L− CD45RO+), and memory central (CD62L+ CD45RO+) T cells subpopulations, along with T cells with TCRαβ, and activated T cells with TCRαβ (CD69+) were all identified. CD4+ T_h_ cells were further subdivided into naïve thymus-negative (CD45RO− CD31−) and positive (CD45RO− CD31+) T_h_ cells.

**Figure 6 ijms-24-11511-f006:**
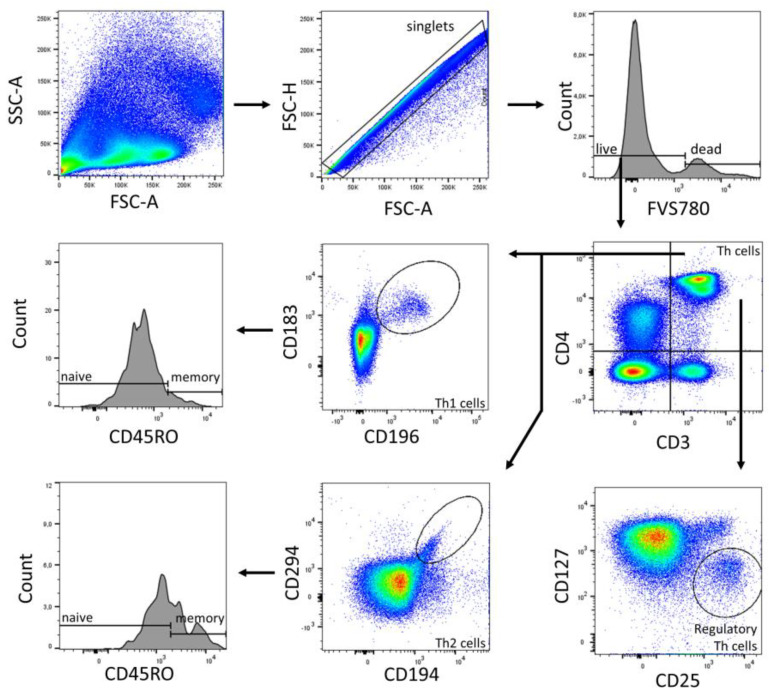
Gating strategy of T cell subpopulations (Panel 2) of a representative CBMC sample. Analysis was performed in FlowJo (BD Biosciences). The gating strategy was based on morphological parameters (FSC-H/FSC-A). FVS 780-negative cells were considered as viable cells. From the viable cells, the number of CD3+ cells was determined. CD4+ T helper cells were identified. Th1 cells (CD183+ CD196+) and Th2 cells (CD194+ CD294+), and their subpopulations naïve (CD45RO−) and memory (CD45RO+) were determined from the T_h_ cell fraction. Regulatory T_h_ cells were identified as CD25+ CD127−.

**Table 1 ijms-24-11511-t001:** Relative amounts of B lymphocyte subpopulations with mean and standard deviation [%]. Group differences were assessed using the Mann–Whitney U test. Test groups were divided into male (*n* = 16) and female (*n* = 21) newborns.

	Sex	Mean ± Standard Deviation [%]	*p*-Value
Innate B cells	Male	6.6 ± 4.0	0.8
(percentage of CD19+ cells)	Female	6.2 ± 3.7	
Naïve B cells	Male	72.0 ± 11.7	0.53
(percentage of CD19+ cells)	Female	75.3 ± 8.3	
Marginal zone memory B cells	Male	1.7 ± 2.1	0.01
(percentage of CD19+ cells)	Female	0.8 ± 1.2	
Class-switched memory B cells	Male	5.7 ± 4.1	0.84
(percentage of CD19 + cells)	Female	5.3 ± 2.6	
Late memory B cells	Male	3.2 ± 2.7	0.073
(percentage of CD19+ cells)	Female	1.7 ± 1.4	
Plasmablasts	Male	5.5 ± 4.0	0.84
(percentage of CD19+ cells)	Female	5.2 ± 2.6	
Transitonal B cells	Male	82.7 ± 11.0	0.89
(percentage of CD19+ cells)	Female	85.4 ± 6.7	
B1 cells	Male	3.7 ± 2.9	0.023
(percentage of CD20+ cells)	Female	1.8 ± 1.5	
B2 cells	Male	0.3 ± 0.3	0.13
(percentage of CD20+ cells)	Female	0.2 ± 0.3	

**Table 2 ijms-24-11511-t002:** Relative amounts of T lymphocyte subpopulations with the mean and standard deviation [%]. Group differences were assessed using the Mann–Whitney U test.

	n	Sex	Mean ± Standard Deviation [%]	*p*-Value
T_h_ cells	16	Male	74.4 ± 6.6	0.22
(percentage of CD3+ cells)	21	Female	76.8 ± 7.2	
Cytotoxic T cells	16	Male	27.9 ± 7.6	0.12
(percentage of CD3+ cells)	21	Female	24.2 ± 7.7	
T_h_ cells with αβ-TCR CD4+ CD3+	16	Male	76.0 ± 6.4	0.5
(percentage of TCR αβ + cells)	21	Female	77.3 ± 7.1	
Activated cytotoxic T_h_-cells with αβ-TCR	16	Male	0.6 ± 0.6	0.27
(percentage of TCR αβ + cells)	21	Female	0.9 ± 0.8	
Memory effector T_h_ cells	16	Male	49.5 ± 13.6	0.48
(percentage of CD3+ CD4+ cells)	21	Female	52.1 ± 15.8	
Memory central T_h_ cells	16	Male	5.7 ± 2.8	0.58
(percentage of CD3+ CD4+ cells)	21	Female	6.7 ± 4.1	
Naïve effector T_h_ cells	16	Male	40.9 ± 15.2	0.17
(percentage of CD3+ CD4+ cells)	21	Female	34.9 ± 11.8	
Naïve central T_h_ cells	16	Male	3.9 ± 1.4	0.62
(percentage of CD3+ CD4+ cells)	21	Female	6.4 ± 9.4	
Memory effector cytotoxic T cells	16	Male	50.8 ± 13.8	0.4
( percentage of CD3+ CD8+ cells)	21	Female	53.7 ± 14.1	
Memory central cytotoxic T cells	16	Male	6.2 ± 4.3	0.66
(percentage of CD3+ CD8+ cells)	21	Female	7.8 ± 6.4	
Naïve effector cytotoxic T cells	16	Male	39.4 ± 14.6	0.24
( percentage of CD3+ CD8+ cells)	21	Female	34.1 ±1 0.6	
Naïve central cytotoxic T cells	16	Male	3.6 ± 1.6	0.73
(percentage of CD3+ CD8+ cells)	21	Female	4.5 ± 4.1	
Naïve thymus negative T_h_ cells	16	Male	24.4 ± 9.8	0.005
(percentage of CD3+ CD4+ cells)	21	Female	15.9 ± 5.7	
Naïve thymus-positive T_h_ cells CD31+	16	Male	65.7 ± 9.4	0.055
(percentage of CD3+ CD4+ cells)	21	Female	70.8 ± 9.6	
T_h_1 cells	11	Male	2.8 ± 1.6	0.61
(percentage of CD3+ cells)	16	Female	2.8 ± 1.9	
Naïve T_h_1 cells	11	Male	87.2 ± 7.0	0.54
(percentage of CD3+ CD4+ CD183+ CD196+ cells)	16	Female	88.1 ± 6.8	
Memory T_h_1 cells	11	Male	12.8 ± 7.0	0.51
(percentage of CD3+ CD4+ CD183+ CD196+ cells)	16	Female	11.9 ± 6.8	
T_h_2 cells	11	Male	0.7 ± 0.5	0.54
(percentage of number of CD3+ cells)	16	Female	0.5 ± 0.3	
Naïve T_h_2 cells	11	Male	66.8 ± 8.6	0.9
(percentage of CD3+ CD4+ CD294+ CD194+ cells)	16	Female	67.6 ± 11.2	
Memory T_h_2 cells	11	Male	33.2 ± 8.6	0.9
(percentage of CD3+ CD4+ CD294+ CD194+ cells)	16	Female	32.4 ± 11.2	
Regulatory T cells	11	Male	2.3 ± 0.6	0.8
(percentage of CD3+ cells)	16	Female	2.4 ± 0.7	

**Table 3 ijms-24-11511-t003:** Definition of B and T lymphocyte subpopulations according to surface markers.

Lymphocyte Subsets		Definition
B lymphocytes	B cells	CD19+
	Innate B cells	CD19+ CD27− IgD− IgM−
	Naïve B cells	CD19+ CD27− IgD+ IgM+
	Memory B1 cells	CD19+ CD27+
	Marginal zone memory B cells	CD19+ CD27+ IgD+ IgM+
	Class-switched memory B cells	CD19+ CD27+ IgD− IgM−
	Late memory B cells	CD19+ CD27+ CD38+ IgM+
	Plasmablasts	CD19+ CD27+ CD38++ IgM−
	Transitional B cells	CD19+ CD20+ CD27− CD38+
	B1 cells	CD20+ CD27+ CD43+
	B2 cells	CD20+ CD27+ CD43−
T lymphocytes	T cells	CD3+
	T helper cells	CD3+ CD4+
	Cytotoxic T cells	CD3+ CD8+
	T helper cells with αβ-TCR	TCRαβ+ CD4+
	Cytotoxic T helper cells with αβ-TCR	TCRαβ+ CD8+
	Activated T helper cells αβ-TCR	TCRαβ+ CD4+ CD69+
	Activated cytotoxic T helper cells with αβ-TCR	TCRαβ+ CD8+ CD69+
	Memory effector T helper cells	CD3+ CD4+ CD62L− CD45RO+
	Memory central T helper cells	CD3+ CD4+ CD62L+ CD45RO+
	Naive effector T helper cells	CD3+ CD4+ CD62L− CD45RO−
	Naive central T helper cells	CD3+ CD4+ CD62L+ CD45RO−
	Memory effector cytotoxic T cells	CD3+ CD8+ CD62L− CD45RO+
	Memory central cytotoxic T cells	CD3+ CD8+ CD62L+ CD45RO+
	Naïve effector cytotoxic T cells	CD3+ CD8+ CD62L− CD45RO−
	Naïve central cytotoxic T cells	CD3+ CD8+ CD62L+ CD45RO−
	Naïve thymus negative T helper cells	CD3+ CD4+ CD31− CD45RO−
	Naïve thymus positive T helper cells	CD3+ CD4+ CD31+ CD45RO−
	T helper cells 1	CD3+ CD4+ CD183+ CD196+
	Naive T helper cells 1	CD3+ CD4+ CD183+ CD196+ CD45RO−
	Memory T helper cells 1	CD3+ CD4+ CD183+ CD196+ CD45RO+
	T helper cells 2	CD3+ CD4+ CD194+ CD294+
	Naïve T helper cells 2	CD3+ CD4 + CD45RO− CD194+ CD294+
	Memory T helper cells 2	CD3+ CD4+ CD45RO+ CD194+ CD294+
	Regulatory T cells	CD3+ CD4+ CD25+ CD127−

**Table 4 ijms-24-11511-t004:** Antibodies used for immunophenotyping.

Marker	Fluorophore	Clone	Vendor	Catalog	Panel
CD127	Alexa-Fluor 647	HIL-7R-M21	BD	55,8598	T cells-panel 2
CD183	BV480	CXCR3	BD	74,6283	T cells-panel 2
CD19	APC-R700	HIB19	BD	56,4977	B cells-panel 1
CD194 (CCR4)	PE	1G1	BD	55,1120	T cells-panel 2
CD196 (CCR6)	APC-R700	CCR6	BD	56,5173	T cells-panel 2
CD20	BB700	2H7	BD	74,5889	B cells-panel 1
CD21	BV421	B-Ly4	BD	56,2966	B cells-panel 1
CD25	BV421	M-A251	BD	56,2442	T cells-panel 2
CD27	APC	L128	BD	33,7169	B cells-panel 1
CD294 (CRTH2)	BV650	BM16	BD	74,0616	T cells-panel 2
CD3	BV786	SK7	BD	56,3800	T cells-panel 1 T cells-panel 2
CD31	BV421	L133.1	BD	74,4801	T cells-panel 1
CD38	PE	HB-7	BD	34,5806	B cells-panel 1
CD4	PE-CF594	SK3	BD	56,6317	T cells-panel 1 T cells-panel 2
CD43	BV605	1G10	BD	56,3378	B cells-panel 1
CD45RO	BV605	UCHL1	BD	56,2791	T cells-panel 1 T cells-panel 2
CD5	BV650	L17F12	BD	74,2551	T cells-panel 1
CD62L	BB700	SK11	BD	74,5995	T cells-panel 1
CD69	BV480	FN50	BD	74,7519	T cells-panel 1
CD8	APC-R700	SK1	BD	56,5192	T cells-panel 1
IgD	PE-CF594	IA6-2	BD	56,2540	B cells-panel 1
IgM	BV480	G20-127	BD	56,6146	B cells-panel 1
TCRαβ	FITC	WT31	BD	33,3140	T cells-panel 1
TCRγδ	PE	11F2	BD	33,3141	T cells-panel 1

## Data Availability

The data presented in this study are available on request from the corresponding author.

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
