# Peer review of "Sex Differences in the Frequencies of B and T Cell Subpopulations of Human Cord Blood"

_ijms, 2023, doi:10.3390/ijms241411511_

Round 1

Reviewer 1 Report

In the current manuscript entitled “Sex differences in frequencies of B and T cell subpopulations of human cord blood” the author explored how the sex of the newborn affects the lymphocytes subpopulation in cord blood. However, the rational of this study is not well descripted.

Major comments:

What is the clinical importance to explore the immune phenotype in cord blood as it would be have changed afterbirth with exposure of numerous environmental factors. Also, most of the studied subpopulation are not terminally differentiated so it could have changed as they expose with different environment conditions afterbirth. How will this initial immune phenotyping hold later in their life?

Minor comment:

1.     Also, it would be important to show the characteristic of newborns.

2.     Gating strategy for individual populations should be shown.

There are few minor errors.

Author Response

  • What is the clinical importance to explore the immune phenotype in cord blood, as it changes afterbirths with exposure of numerous environmental factors?

Thank you for your comment, we have adapted and revised the section and added the necessary background information (ll. 51-73).

  • Most of the studied subpopulation are not terminally differentiated so it could have changed as they expose with different environment conditions afterbirth. How will this initial immune phenotyping hold later in their life?

Thank you for the feedback. We have provided a more detailed explanation (ll.57-63).

  • Characteristics of Newborns

Thank you for your comments. We have addressed this issue in the manuscript (ll. 67-73).

  • Gating strategy for individual populations should be shown

Thank you for your feedback. We have revised the methodology section.

Reviewer 2 Report

The research paper investigates the potential impact of newborn sex on lymphocyte subpopulations in cord blood, aiming to shed light on the intricate processes of immune development during early life. Using flow cytometry, the study characterizes B and T cell subpopulations in cord blood samples from male and female newborns. The results reveal sex differences in B and T cell counts, with higher B cell counts observed in subsets of cord blood from male newborns and higher T cell counts in subsets from female newborns. The study highlights the importance of considering sex as a potential influencing factor in immune system development, emphasizing the need for further research in this area.

Revision Comments:

1.     The introduction should provide a more comprehensive background by explaining the importance of immune development during the perinatal period and the potential influence of newborn sex on lymphocyte subpopulations.

2.     Clearly state the specific research question and objectives of the study in the introduction to provide a clear focus for the readers.

3.     Provide a detailed description of the flow cytometry methodology, including the specific antibodies used, gating strategies, and quality control measures, to ensure reproducibility and allow other researchers to replicate the study.

4.     Organize the results section in a structured manner, presenting the findings for B and T cell subpopulations separately and providing statistical analyses to support the observed differences between male and female cord blood samples.

5.     Use appropriate statistical tests to determine the significance of the observed differences and clearly report the p-values or confidence intervals.

6.     Discuss the potential biological mechanisms underlying the sex differences in cord blood lymphocyte subpopulations, such as hormonal influences or genetic factors, to provide a more comprehensive understanding of the findings.

7.     Compare the study's results with previous literature on sex-related differences in immune development to highlight the novelty and contribution of the current study.

8.     Discuss the potential clinical implications of the findings, such as the relevance of these sex differences for disease susceptibility, vaccine responses, and personalized medicine approaches.

9.     Address the limitations of the study, such as the relatively small sample size, and discuss how these limitations may impact the generalizability of the findings.

10.  Provide a clear roadmap for future research by suggesting specific directions for further investigation, such as exploring the long-term consequences of the observed sex differences or investigating the underlying molecular mechanisms.

11.  Consider the potential influence of confounding factors, such as maternal factors or gestational age, and discuss how these factors were addressed or controlled for in the study.

12.  Include a discussion on the potential implications of the findings for prenatal and neonatal healthcare practices, highlighting how this knowledge may inform interventions or treatments.

13.  Incorporate a section on the clinical relevance of the observed sex differences in cord blood lymphocyte subpopulations, discussing how this information may be applied to improve disease diagnosis, prognosis, or treatment strategies.

14.  Provide a balanced perspective by acknowledging alternative explanations or interpretations of the findings and discussing their potential implications.

15.  Improve the overall readability and clarity of the article by reorganizing the sections, ensuring a logical flow of information, and using clear and concise language throughout.

Proofread the article for grammatical and stylistic errors, ensuring that the writing is clear, concise, and consistent throughout.

Author Response

Reviewer 2

2.1. The introduction should provide a more comprehensive background by explaining the importance of immune development during the perinatal period and the potential influence of newborn sex on lymphocyte subpopulations.

Thank you for your comments. We have addressed this issue in the manuscript as follows. We kindly refer to ll. 51-73 and 110-151 in our manuscript.

  • Clearly state the specific research question and objectives of the study in the introduction to provide a clear focus for the readers.

We thank you for your comment. We clarified our research questions and objectives.

  • Provide a detailed description of the flow cytometry methodology, including the specific antibodies used, gating strategies and quality control measures to ensure reproductibility and allow other researchers to replicate the study.

Thank you for your feedback. We have revised the material and methods, we kindly refer to ll. 394-488 in our methodology section.

  • Organize the results section in a structured manner presenting the finding for B and T cell subpopulations separately and providing statistical analyses to support the observed differences between male and female cord blood samples. Could you please reiterate how the results section should be restructured, as we already have separate subsections for B-cell and T-cell results in the analysis?

We thank for this annotation and tried to implement your propositions into our manuscript (please see results section).

  • Use appropriate statistical tests to determine the significance of the observed differences and clearly report the p-values or confidence intervals.

Thank you for the suggestion, we have revised this section (ll. 481-488). We specified that quantitative data analysis and the p-values.

.

  • Discuss the potential biological mechanisms underlying the sex differences in cord blood lymphocyte subpopulations such as hormonal influence genetic factors to provide a more comprehensive understanding.

We thank the reviewer for this important annotation and specified this issue in our manuscript. We focussed on two main factors : the influence of sex hormones and differences in the number of immune related genes on the X chromosome (ll.226-322).

  • Compare the study`s results with previous literature on sex related differences in immune development to highlight the novelty and contribution of the current study.

Thank you for the comment.  As mentioned in the manuscript, the available data on sex-specific differences in cord blood lymphocyte subpopulations are limited. There are no reference values in the literature for lymphocyte subpopulations labelled with “our” surface markers. Therefore, our findings represent a novelty and provide a basis for further research to establish reliable sex-specific reference values.

  • Discuss the potential clinical implications of the findings such as the relevance of these sex differences for disease susceptibility, vaccine responses and personalized medicine approaches.

Thank you for your comments. We have addressed this issue and we discussed possible improvement in prenatal diagnostics, vaccination schedules and tailored therapeutic approaches (ll.354-360).

  • Address the limitations of the study such as the relatively small sample size and discuss how these limitations may impact the generalizability of the findings.

Thank you for the suggestion. We added limitations to our manuscript, such as the small cohort size potentially restricting the generalizability of our study.

  • Provide a clear roadmap for further investigation such as exploring the long-term consequences of the observed sex differences or investigating the underlying molecular mechanisms.

We thank the reviewer for this idea, we provided a more detailed roadmap in ll. 372-378.

  • Consider the potential influence of confounding factors, such as maternal factors or gestational age, and discuss how these factors were addressed or controlled for in the study.

Thank you for the suggestion - we have revised the passage. This is an important point – potential confounders must be considered (please see ll.365-371).

  • Include a discussion on the potential implications of the findings for prenatal and neonatal healthcare practices highlighting how this knowledge may inform intervention or treatments.

Thank you for the input. We have addressed comments 8, 12, and 13 in one section. We kindly refer to ll. 354-378 in our manuscript.

  • Incorporate a section on the clinical relevance of the observed sex differences in cord blood lymphocyte subpopulation, discussing how this information may be applied to improve disease diagnosis, prognosis or treatment strategies.

Thank you for the input. We have addressed comments 2.8, 2.12, and 2.13 in one section. . We kindly refer to ll. 354-378 in our manuscript.

  • Provide a balanced prespective by acknowledging alternative explanations or interpretations of the findings and discussing their potential implications.

Thank you for your input. We have addressed comments 2.11 and 2.14 in one section. Please refer to section 2.11 for further information.

  • Improve the overall readability and clarity of the article by recognizing the sections, ensuring a logical flow of information and using clear and concise language throughout.

Thank you for your feedback. We have revised the manuscript and highlighted the modified sections in yellow colour.

Round 2

Reviewer 1 Report

The authors discussed all concerns. 

Not applicable

Reviewer 2 Report

The authors revised the manuscript according to the comments, and my recommendation is accepted in its current form.